# Identification of the Rare Ala871Glu Mutation in the Androgen Receptor Gene Leading to Complete Androgen Insensitivity Syndrome in an Adolescent Girl with Primary Amenorrhea

**DOI:** 10.3390/children9121900

**Published:** 2022-12-03

**Authors:** Aikaterini Kapama, Dimitrios T. Papadimitriou, George Mastorakos, Nikolaos F. Vlahos, Maria Papagianni

**Affiliations:** 1Department of Internal Medicine, 401 General Military Hospital of Athens, 11525 Athens, Greece; 2Unit of Endocrinology, Diabetes Mellitus and Metabolism, Second Department of Obstetrics and Gynecology, Aretaieion University Hospital, National and Kapodistrian University of Athens, 12462 Athens, Greece; 3Department of Pediatric-Adolescent Endocrinology and Diabetes, Athens Medical Center, 15125 Marousi, Greece; 4Second Department of Obstetrics and Gynecology, Aretaieion University Hospital, National and Kapodistrian University of Athens, 12462 Athens, Greece; 5Department of Nutrition and Dietetics, University of Thessaly, 42132 Trikala, Greece; 6Unit of Endocrinology, Diabetes and Metabolism, Third Department of Pediatrics, Aristotle University of Thessaloniki, Hippokrateion Hospital of Thessaloniki, 54642 Thessaloniki, Greece

**Keywords:** complete androgen insensitivity syndrome, androgen receptor, mutation, hormone replacement therapy, CAIS, PAIS

## Abstract

Complete Androgen Insensitivity Syndrome (CAIS) is a rare genetic condition by mutations in the androgen receptor (AR) gene resulting in target issue resistance to androgens and a female phenotype in genetically male individuals. A 16-year-old phenotypically female individual presented to our clinic with primary amenorrhea. Her clinical evaluation showed normal female external genitalia, Tanner III breast development and sparse pubic and axillary hair (Tanner stage II). Hormonal assessment revealed increased concentrations of Luteinizing Hormone (LH), Testosterone and Antimüllerian Hormone (AMH). Image studies detected no uterus or gonads, but a blind vagina and the karyotype was 46, XY. These findings suggested the diagnosis of CAIS, and genetic testing of the AR gene revealed a rare pathogenic mutation of cytosine to adenine (c.2612C>A) replacing alanine with glutamic acid at position 871 (p.Ala871Glu) in the AR, previously described once in two adult sisters. The patient underwent gonadectomy and received hormonal replacement therapy. This study expands the AR mutation database and shows the complexity and the importance of prompt diagnosis, proper management, and follow-up for CAIS patients, underlining the need for standardized protocols.

## 1. Introduction

Morris first described Androgen Insensitivity Syndrome (AIS) as a clinical entity characterized by end-organ resistance to androgens. The pathogenetic mechanism is a defect in the androgen receptor (AR) gene, which encodes the AR, found on the X chromosome (Xq12). The syndrome is inherited in an X-recessive manner, and 30% of the AR mutations are de novo [1].

The AR gene-coding region is composed of a 920-amino acid sequence, organized in eight exons according to the National Center for Biotechnology Information (NCBI) reference sequence. The AR (NR3C4 gene; Nuclear Receptor subfamily 3, group C, gene 4) belongs to the family of ligand-dependent nuclear hormone receptors, its ligands being testosterone (T) and dihydrotestosterone (DHT). The receptor acts as a transcriptional factor that regulates gene expression in target tissues and has a structural organization composed of 4 different domains: an NH_2_-Terminal Domain (NTD) encoded by exon 1 which is essential for the transcription activation, a DNA-Binding Domain (DBD) encoded by exons 2–3 which interacts with androgen response elements (ARE) in the DNA and is responsible for the dimerization of the AR, a Hinge Region (HR) which contains the nuclear localization sign (NLS) and a Ligand Binding Domain (LBD) encoded by exons 4–8, which interacts with the agonists [2,3].

Many variants within the AR gene have been identified as pathogenic mutations, affecting the AR protein quantitatively or qualitatively. Depending on the degree of the remaining AR function, the syndrome encompasses three medical conditions: Complete Androgen Insensitivity Syndrome (CAIS), Partial Androgen Insensitivity Syndrome (PAIS), and Mild Androgen Insensitivity Syndrome (MAIS) [4]. CAIS is characterized by a female phenotype with normal development of female secondary characteristics, female external genitalia, absent or sparse pubic and axillary hair but absent Müllerian structures and ovaries in a 46, XY individual. The gonads are normal testicles found in the abdomen, inguinal canal, or labio-scrotal region. They normally produce testosterone and its peripheral aromatization to estrogens leads to the initiation of puberty and the development of female secondary characteristics [5]. Primary amenorrhea with a family history of an infertile tall maternal aunt often leads to the diagnosis. In PAIS, the phenotype can vary from hypospadias and genital ambiguity to nearly normal female genital morphology with variable degrees of clitoromegaly. In MAIS, gynecomastia and/or unexplained male infertility are the mere complaints in an otherwise normally virilized man [6].

Currently, there are over nine hundred AIS-related mutations in the AR gene. Four different types of mutations have been discovered to cause defective AR: (I) single point mutations resulting in amino acid substitutions or premature stop codons; (II) nucleotide insertions or deletions most often resulting in alteration of the reading frame; (III) total or partial gene deletions; and (IV) splice-site mutations. However, single-point mutations are more often identified in patients with AIS. The total number of NTD mutations are only 25% of all mutations in AIS patients, most leading to CAIS. In the DBD region, at least 146 mutations have been reported, mainly single-base substitutions. There is a preponderance of mutations in the LBD region, mainly single-base substitutions, representing approximately 2/3 of AR mutations (692 variants), more than half causing the CAIS phenotype [7].

AIS is a rare genetic disorder with a prevalence of 2–5:100,000 according to the National Institute of Health (NIH) and an incidence of 1:20,000–1:90,000 male births [8,9]. AIS is classified as a disorder of sex development (DSD) reported as the most frequent type of 46, XY DSD followed by gonadal dysgenesis [10,11,12,13].

## 2. Case Report

### 2.1. Patient Information

A 16-year-old female presented to our clinic with primary amenorrhea. She was the only child of non-consanguineous healthy parents and was born full term after non-complicated pregnancy by normal vaginal delivery with a birth weight of 3.4 kg. She was a champion athlete in swimming and initiated puberty (thelarche and adrenarche) late at the age of 13 years, with no history of delayed puberty and/or growth in her parents, however. Personal and family medical history was uneventful, and she had never had a diagnostic workup before.

### 2.2. Clinical Findings and Diagnostic Assessment

Her height was 173 cm (between 90th and 97th centile), her weight was 57 kg (50th centile) and her body mass index was 19 kg/m^2^ (25th centile). Evaluation of her growth charts revealed linear growth consistent with her target height until 13.5 yrs (Figure 1). Delayed pubertal development and bone maturation by 1 year at 16.3 yrs (evaluated by BoneXpert ver. 3.2.1, © Visiana, Hørsholm, Denmark) led to a higher near adult height 174 cm compared to her target height 168.5 cm (Figure 1). Her body proportions: arm spam, upper to lower segment ratio, and sitting height were normal.

On examination, she was normotensive. There were no dysmorphic features. Her breast development was Tanner stage III, her pubic hair was Tanner stage II (sparse fine hair on pubic area), and her axillary hair was Tanner stage II (sparse fine hair). She had normal female external genitalia with no signs of virilization.

Hormonal evaluation at 08:00 h showed luteinizing hormone (LH), testosterone (T), dihydrotestosterone (DHT) and Antimüllerian hormone (AMH) above the normal range for her bone age and Tanner stage III (https://www.mayocliniclabs.com/test-catalog/ (accessed on 7 May 2021)) with normal sex hormone binding globulin (SHBG) and vitamin D insufficiency (26.3 ng/mL) (Table 1).

Taking into consideration the increased testosterone concentrations combined with the absence of virilization signs, a transabdominal pelvic ultrasound and a karyotype were requested. The pelvic ultrasound could not find any uterine structure or gonads of any kind, nor did a subsequent MRI with i.v. contrast agent, which revealed however a blind vagina measuring 8 cm in length. Considering the clinical picture, the hormonal results, and the absence of internal female genitalia, CAIS was suspected, the later result of a 46, XY karyotype being indicative of the diagnosis. Therefore, genetic testing with next-generation sequencing (NGS) of the AR gene was ordered. Genomic DNA was extracted from the peripheral blood lymphocytes of the patient. The AR reference sequences were amplified using the Ion AmpliSeq^TM^ Exome RDY Kit (ThermoFisher Scientific (Waltham, MA, USA)) and sequenced using the Ion Chef Instrument and the Ion GeneStudio^TM^ S5 System Alamut Visual and Varsome Clinical (Saphetor (Lausanne, Switzerland)). The reference gene was UCSC hg19. Sequencing analysis of all the eight exons revealed a novel hemizygous mutation of cytosine to adenine (c.2612C>A) on exon eight of the LBD, replacing alanine with glutamic acid at position 871 (p.Ala871Glu).

The mutation was characterized as likely pathogenic according to the ACMG/AMP guidelines using in silico predictive programs [14]. Information concerning the three-dimensional structure of the AR protein and its interactions with agonists were obtained from the RSCB PDB database (http://www.rcsb.org/ accessed on 10 September 2022) using PDB ID: 2ama structure as a template. We also developed a model of a 3D structure of the mutated protein using UCSF Chimera open-source software [15]. The Ala871Glu mutation lies within the helix 11 of the LBD, a region that has been associated not only with AR specificity but also with interdomain and co-activator interactions [16]. Alanine is a hydrophobic amino acid with a short non-polar side chain, while glutamic acid is a hydrophilic amino acid with a larger negatively charged side chain. While residue 871 does not directly interact with the bound ligand, it resides near the Met743 amino acid, one of the 18 amino acids that form the ligand-binding pocket, residues which interact more or less directly with the ligand [17]. Regarding the substitution of the hydrophobic alanine by the hydrophilic glutamic acid, it can be noted that this mutation changes the hydrophobicity profile (Figure 2). In addition, analyzing the substitution on a 3D model structure, we found that the distance between the Glu871 and Met743 is 3.03 Å in contrast with the 4.06 Å between Ala871 and Met743 (Figure 3).

### 2.3. Therapeutic Intervention and Outcome

Based on her medical condition, the patient was referred for diagnostic laparoscopy.

Tumor markers, performed to exclude the possibility of malignancy from testicular remnants, were negative in the preoperative evaluation. The laparoscopy revealed—as expected—two hypoplastic gonads which were then removed, histologically confirmed as testicles.

The histopathological examination showed two ovoid structures measured 4.7 × 3 × 1.7 cm and 3.8 × 2 × 1.2 cm. Both had a superficial cystic lesion of 1.3 cm and 0.8 cm, respectively. The microscopic image showed two hypoplastic testicles consisting of seminiferous tubules with 80% hyperplastic Sertoli cells and some spermatogonia. Leydig cells were also present composing 20% of the total volume. Both structures were covered by a fibrous capsule which was regionally continued with parts of fallopian tubes. No signs of malignancy were found.

Following gonadectomy, the patient was put on hormone replacement therapy (HRT), initially with 2 mg/day oral valerate estradiol (she initially preferred to take the medication *per os* and not transdermal as she was swimming 2–3 h/day) Within a month, testosterone returned to the normal female range with markedly increased gonadotropins, as expected for CAIS [5], with estradiol still low despite patient’s compliance. The use of an estradiol patch instead (estradiol hemihydrate 100 mg twice a week) clearly improved the hormonal profile. The patient underwent bone mass measurement of her lumbar spine by dual-energy X-ray absorptiometry (DEXA) five months after gonadectomy while on 2 mg/day oral valerate estradiol for the first month and estradiol patch (estradiol hemihydrate 100 mg twice a week) thereafter. Her lumbar spine bone mineral density (BMD) was 0.814 g/cm^2^ with a Z-score of −1.5 in the range of osteopenia according to her bone age, despite optimal vitamin D repletion following the initial work-up [18].

## 3. Discussion

CAIS is defined as the total absence of AR function in a 46, XY individual. Clinically, CAIS is expressed as a female phenotype in a genetically male individual, with the initial presentation varying according to the age of diagnosis [9]. The syndrome can be diagnosed in neonatal life and early infancy, in adolescence or even in adulthood [19]. In neonates, it can be diagnosed because of incongruence between genotype and phenotype, in cases where amniocentesis had been performed. In neonates but also in older children, CAIS should be suspected in phenotypical females with bilateral inguinal hernia or apparent labial swellings due to undescended testis [20,21]. In adolescence, primary amenorrhea, or delayed puberty with or without inguinal hernia is the most common clinical presentation. In the present case, the subject had normal albeit delayed breast development (Tanner stage III), sparse fine axillary and pubic hair and a blind vagina pouch of adequate length [20,22]. Testicles are always present in CAIS and may be positioned in the abdominal region, as in this case [23]. The older the patient, the more difficult it gets to identify them radiologically, due to the ongoing testicular atrophy, needing a research laparoscopy for identification and removal of the hypoplastic testes. Wolffian duct derivatives (epididymis, vas deferens)—not found in this patient—may be present in some cases [24,25], and regression of the Müllerian structures can be incomplete [26,27,28], as it is the case.

Regarding adult height, CAIS patients are usually taller than their female target values, inferring a role for the Y chromosome, as in this case [5]. As previously shown, bone age maturation follows the female pattern and is delayed, consistent with the concomitant pubertal delay [5], and this was also the case in our patient. However, the presence of the Y chromosome in CAIS patients increases the risk for malignant testicular germ cell tumors (TGCT) as in other types of DSD, justifying thus, recommendation of early prophylactic gonadectomy [29]. In a recent systematic review including 456 CAIS patients a percentage of 6.14% had premalignant lesions, the majority (81.4%) after 12 years of age, with adult patients diagnosed with CAIS presenting a 1.3% rate of malignancies in [30]. Similarly, in a recent series, 1 out of 35 CAIS patients had a Leydig cell tumor [31]. Nevertheless, it has been proposed that gonadectomy can be delayed allowing automatic initiation of puberty. This strategy is strongly debated not only for the increased risk at least for in situ malignancy but also for the psychosocial implications of a late gonadectomy in a fully conscious adolescent who must first undergo a thorough grinding preparation in order to understand, accept and fully consent to the procedures planned [32]. In the present patient gonadectomy was performed after initiation of puberty, as diagnosis was made before primary amenorrhea and luckily, the histological examination of the testicles showed no signs of malignancy.

Patients with CAIS run an increased risk of low bone mineral density (BMD) mainly at the lumbar spine [33,34,35] but also at the femoral neck, [34,36,37] despite estrogen replacement therapy, and also even before gonadectomy—as it is the case with this patient [34,38,39], who initiated female puberty based on aromatized testosterone alone. The latter seems unable to normally suppress gonadotrophins, meaning that spontaneous puberty in CAIS patients is achieved but with lower estradiol concentrations than needed for optimal pubertal growth and maturation.

Hormonal profile in CAIS patients depends on their age. In 46, XY neonates with normal female external genitalia increased concentrations of basal LH and testosterone, during the postnatal mini-puberty surge [40] are missing [5], indicative of CAIS. This lack of the traditional male ‘mini-puberty’ of neonatal life suggests a ‘feminized’ pattern in the activation of the gonadotropic axis underlying the importance of a functional AR for pubertal programming [5], expecting the onset of puberty to follow a female than a male pattern. Later, in infancy and childhood, outside the optimal diagnostic window of male mini-puberty, stimulation with human chorionic gonadotropin (hCG) and luteinizing hormone-releasing hormone (LHRH) may show increased concentrations of testosterone and LH, respectively. However, these findings are not universal and while adequate testosterone response to hCG may exclude biosynthetic defects in testosterone synthesis, an inadequate response does not necessarily exclude CAIS. Similarly, in late childhood LH concentrations may not increase enough, even after an LHRH stimulation test, underlying the importance of timely referral and diagnosis [41]. In adult CAIS patients, testosterone, DHT and androstenedione (Δ4) concentrations are usually in the normal range for men while estradiol concentrations are low for females and within the normal reference range for males. LH is usually increased and FSH within the normal range. T/DHT ratio may be increased implying a secondary 5a-reductase 2 deficiency [20,42]. AMH, which is produced by Sertoli cells in normal males, is increased after the onset of puberty in CAIS patients. This is probably because in CAIS the negative regulation of AMH by testosterone is missing, leading to increased concentrations of AMH as in defects in androgen synthesis and/or action [43,44]. Our patient had hormonal measurements compatible to the CAIS pubertal profile reported by Papadimitriou et al. [5] with testosterone and DHT concentrations within the normal male range, with normal T/DHT ratio, Δ4 and estradiol concentrations within the normal female range for pubertal stage Tanner III and increased LH and AMH with normal FSH concentrations.

Radiological examinations are a useful tool for the possible identification of Müllerian structures, the localization of the testicles, and the exploration of possible underlying malignancies. The absence of Müllerian duct derivatives can be verified by ultrasonography, but Magnetic Resonance Imaging (MRI) is the study of choice for identification of uterovaginal abnormalities [45]. While transabdominal ultrasonography is a useful initial tool for the localization of undescended or non-palpable testes in CAIS patients, its accuracy is limited when the testes are above the inguinal ring and probably atrophic [46,47]. In the cases where the testicles are located high in the inguinal canal or in the abdomen, MRI is more sensitive [47]. In the present patient, a hypoplastic uterus was recognized in both imaging studies but the testicles were not recognized by either ultrasound or MRI due to their atrophy. In adolescent girls with breast development and primary amenorrhea, the presence of the uterus should be assessed. An absent uterus necessitates a chromosomal analysis [48]. A female genotype results in the diagnosis of Müllerian agenesis, while a 46, XY karyotype in a female phenotype with an absent uterus is typical of CAIS [49].

The definite diagnosis of CAIS is proven by the identification of a hemizygous mutation in the AR gene by genetic testing. Approximately 550 loss-of-function mutations throughout the coding region of the AR gene have been described to be causative of CAIS, mostly single-base substitutions in the LBD area (exons 4–8) [7]. Specifically in the 871 coding region mutations that lead either to PAIS or MAIS, two have been described with a mutation identical to that of the present case (Ala871Glu) responsible also for CAIS phenotype (Table 2). A synonymous mutation Ala871Ala has been described by Akcay et al., in a 3-year-old patient with PAIS [50]. Missense mutations replacing alanine with valine (Ala871Val) have been described leading to various PAIS and a MAIS phenotype. Hiort et al., have described a patient with severe hypospadias and bilateral cryptorchidism (PAIS) and a patient with severe hypospadias and micropenis (PAIS) [51,52]. Albers et al., have described a 4-year-old patient with severe hypospadias (PAIS), Zenteno et al., a 24-year-old patient with bilateral gynecomastia (MAIS), Bhangoo et al., a 3-year-old patient with isolated micropenis (PAIS), Audi et al., a 1-year-old patient with hypospadias and pubertal gynecomastia (PAIS) and Su et al., a 3-year-old patient with penoscrotal hypospadias and Wilms tumor (PAIS) [10,53,54,55,56]. Hiort et al., have also described a mutation Ala871Gly in a female patient with virilized external genitalia (PAIS) [52]. Two sisters aged 60 and 56 years old with the clinical diagnosis of CAIS, developed Sertoli cell tumors and underwent molecular genetic testing revealing the Ala871Glu mutation [57]. These two cases and the present case—to the best of our knowledge—are the only Ala871Glu mutations reported so far. Fortunately, in the present patient, apart from mild hyperplasia of Sertoli cells there were no signs of malignancy.

In CAIS, gonadectomy due to the risk for malignancy renders the patient hypogonadal, requiring HRT, according to female physiology [5]. Where automatic pubertal initiation is not possible due to previous gonadectomy, the conventional strategy is using low dose of estrogen replacement therapy in the beginning, with gradual increase over 2–3 years mimicking the female physiologic tempo of puberty [5]. Progestins are not required due to the absence of a uterus [58]. Different estrogen formulations can be used as HRT. However, regarding the best estrogen formulation, delivery method and dosages for CAIS patients, there is lack of evidence-based information. Transdermal estrogens are considered superior in terms of their more physiologic mode of delivery, the lack of first-pass hepatic metabolism, the reduced risk of thromboembolism and their beneficial role in the bone mineral density, although there are several known difficulties in titration and adherence often related to technical modalities of application. Transdermal estradiol has also a high probability of premature detachment, especially in the humid environment of a swimming pool where this patient spent several hours daily [59]. Alternative natural forms of estrogens such as oral 17β-estradiol or valerate estradiol, which is metabolized to 17β-estradiol after hepatic first-pass, are easier to use with daily oral administration. Ethinyl estradiol or conjugated estrogens are less often used. Oral valerate estradiol leads to increased estrone concentrations and total bio-estrogen activity as well as a delayed steady-state condition achievement [60]. Estrogen substitution in CAIS patients relies on the specialist’s experience and the patient’s convenience, adherence, and response to therapy. In the present case, the initial dose of 2 mg/day of estradiol valerate, equivalent to the adult dose, was reevaluated as it seemed inadequate to normalize estradiol concentrations and then, patient was put on estradiol patch (estradiol hemihydrate 100 mg twice a week) optimizing the hormonal profile thereafter. Testosterone, the main androgen secreted by gonads in CAIS patients, exerts its effects through its aromatization to estrogens. In that sense, testosterone could be useful as an HRT in CAIS women. A recent randomized control trial that enrolled 26 CAIS women concluded that testosterone is a safe alternative HRT to estrogen but is superior to estrogen, only in increasing sexual desire [61]. Finally, psychological, psychiatric and psychosocial support of patients should be not subsided as CAIS patients develop more often anxiety and depression disorders [62,63,64]. Regarding cardiovascular risk, limited data also suggests an increased risk for impaired metabolic profile in CAIS patients. Therefore, clinicians should evaluate body fat, lipid and glucose metabolism in older CAIS patients [65].

## 4. Conclusions

This report outlines the complexity and the importance of prompt diagnosis, proper management, and follow-up of CAIS patients, underlying the need for standardized protocols. In adolescents with primary amenorrhea, hormonal analysis indicative of CAIS and diagnostic imaging revealing the absence of Müllerian structures are helpful in suggesting the clinical diagnosis. Then, a karyotype showing a 46, XY female should lead to the analysis of the AR gene. The pathogenicity of a novel variant can be predicted in silico using bioinformatic tools. According to current evidence, while gonadectomy can be postponed until after pubertal onset, this strategy may hide additional risks and psychosocial difficulties, while not proven superior in terms of pubertal development or bone health. Following gonadectomy, proper HRT is mandatory following the normal female pubertal timing. Thorough assessment of bone health with optimization of calcium and vitamin D metabolism is necessary due to the greater risk for osteopenia and osteoporosis. Timely diagnosis and proper management of CAIS patients require close collaboration of a multidisciplinary team of pediatric and adult endocrinologists, gynecologists, clinical geneticists, and psychiatrists. 

## Figures and Tables

**Figure 1 children-09-01900-f001:**
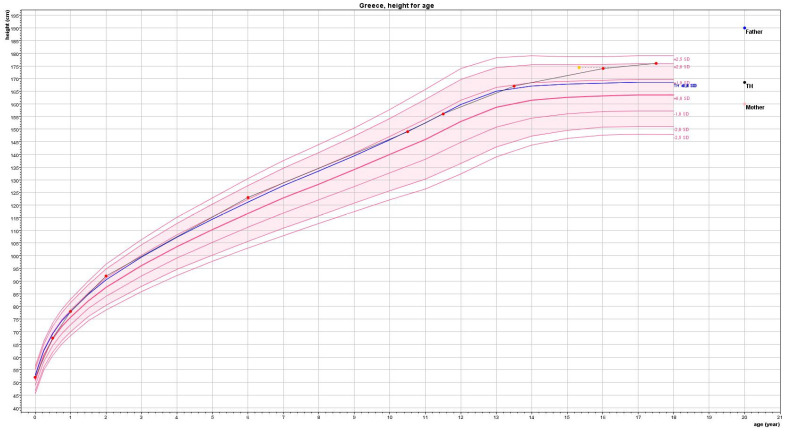
Growth chart; Growth Analyser© 2022–2006, Rotterdam—The Netherlands.

**Figure 2 children-09-01900-f002:**
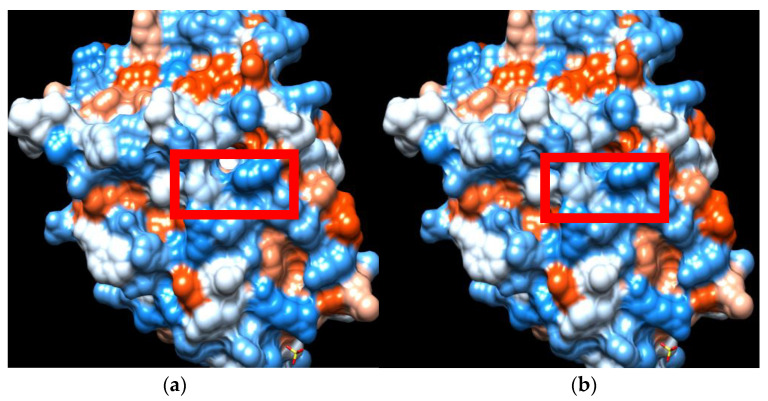
Modification of hydrophobicity between wild type AR (**a**) and Ala871Glu mutant AR (**b**). (Red-hydrophobic region, blue-hydrophilic).

**Figure 3 children-09-01900-f003:**
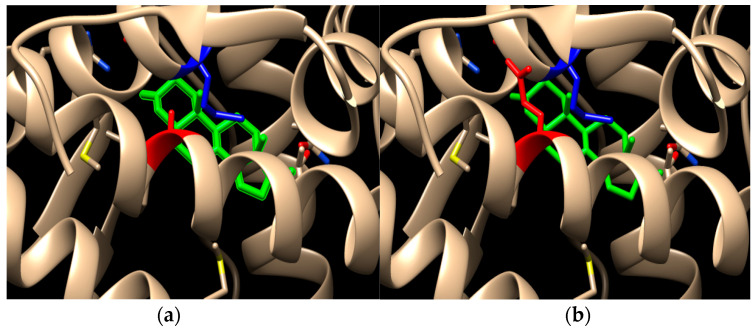
Amino acid changes between wild type AR (**a**) and Ala871Glu mutated protein (**b**) (Red-amino acid 871, blue-Met743, green-DHT).

**Table 1 children-09-01900-t001:** Hormonal evaluation markers at diagnosis and on Hormone Replacement Therapy after gonadectomy.

	At Diagnosis	1 Month after Surgery, on Oral HRT	On Transdermal HRT	Reference Values
FSH (IU/L)	5.6	111.9	2.6	0.1–7.2
LH (IU/L)	18.2	41	0.2	<11.9 (age 11–14) <14.6 (follicular phase <18 yrs)
Estradiol (pg/mL)	30.7	21	476.3	<60
T (ng/dL)	680	14	N/A	17–75
DHT (pg/mL)	400	N/A	N/A	<300
AMH (ng/mL)	10.6	N/A	N/A	0.6–7.8 (age 15–19)
SHBG (nmol/L)	57.2	N/A	N/A	18–87

LH and FSH were measured with an Elecsys immunoassay analyzer (Roche). Testosterone and estradiol were measured by liquid chromatography/tandem mass spectrometry. AMH was measured using the AMH/Mullerian-inhibiting substance ELISA kit (Immunotech-Beckman, Marseilles, France).

**Table 2 children-09-01900-t002:** Androgen receptor gene mutations in 871 codon and their phenotypes.

Phenotype	Age	Mutation	Reference
PAIS (Quigley scale 3)	3	Ala871Ala	[50]
PAIS (hypospadias and cryptorchidism)		Ala871Val	[51]
PAIS (hypospadias and micropenis)		Ala871Val	[52]
PAIS (hypospadias)	4	Ala871Val	[53]
MAIS (bilateral gynecomastia)	24	Ala871Val	[54]
PAIS (isolated micropenis)	3	Ala871Val	[55]
PAIS (hypospadias and gynecomastia)	1	Ala871Val	[10]
PAIS (hypospadias and Wilms tumor)	3	Ala871Val	[56]
PAIS (female with virilized external genitalia)		Ala871Gly	[52]
CAIS (female with Sertoli cell tumor)	60 and 57	Ala871Glu	[57]
CAIS	16	Ala871Glu	This report

## Data Availability

The data presented in this study are available within the article.

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
