# Peer review of "Identification of the Rare Ala871Glu Mutation in the Androgen Receptor Gene Leading to Complete Androgen Insensitivity Syndrome in an Adolescent Girl with Primary Amenorrhea"

_children, 2022, doi:10.3390/children9121900_

Round 1

Reviewer 1 Report

Androgen insensitivity syndrome (AIS) is a common form of 46, XY disorder in sex development disease. It is due to the androgen receptor (AR) gene mutations and includes clinical subgroups of complete AIS (CAIS) and partial AIS (PAIS), along with a vast area of clinical heterogeneity of completely normal female external genitalia to male infertility. In this manuscript, the authors reported a rare AR mutation (Ala871Glu), leading to CAIS. Besides the clinical findings and diagnostic assessment, a three-dimensional (3D) model for the Ala871Glu mutation was constructed. The Ala871Glu mutation lies within the helix 11 of the ligand binding domain (LBD), a region that has been associated not only with AR specificity but also with interdomain and co-activator interactions, the 3D model showed hydrophobicity profile changes between wild type AR and Ala871Glu mutant AR, revealed the potential molecular mechanism of the Ala871Glu of AR induced AIS. Furthermore, the authors outlined the importance of timely diagnosis, optimal therapeutic interventions, proper management, and follow-up of CAIS patients, emphasized the need for standardized protocols. The manuscript was well organized, and the data was valuable. The manuscript can be accepted for publication with minor revisions

1.      In Page5, Figure 3, if it is possible, please label the position of Ala871Glu mutant in the three-dimensional coordinates of the protein structure; In addition, besides the Red-hydrophobic region, blue-hydrophilic, please indicate what the green color presented

In Page 8, line 349, please correct the word “underling “

Author Response

Thank you for giving us the opportunity to submit a revised draft of our manuscript titled “Identification of a rare Androgen Receptor mutation leading to Complete Androgen Insensitivity Syndrome. The importance of timely diagnosis, optimal therapeutic interventions, and multidisciplinary management.” to Children. We appreciate the time and effort that you and the reviewers have dedicated to providing us with your insightful comments and suggestions, which, hopefully, have significantly improved the manuscript.

We have highlighted the changes within the manuscript using the “Track Changes” function, as you suggested.

Here is detailed answer to the comments.

Yours sincerely,

Dimitris T. Papadimitriou and Maria Papagianni, and on behalf of all our co-authors.

Comments from reviewer 1

  • Comment 1:  In Page5, Figure 3, if it is possible, please label the position of Ala871Glu mutant in the three-dimensional coordinates of the protein structure; In addition, besides the Red-hydrophobic region, blue-hydrophilic, please indicate what the green color presented

Response: Thank you for your comment. We have corrected the label of the figure 3 and added the proper explanation, indicating what each color means.

  • Comment 2: In Page 8, line 349, please correct the word “underling “

Response: Thank you for your comment. We corrected it.

Reviewer 2 Report

Dear colleagues,

Thank you very much for submitting your manuscript. Primary amenorrhea is a pathology that requires multidisciplinary evaluation, with significant implications for medical practice. The topic is relevant and exciting to the field of the journal. The text is clear and easy to read.

The title needs to change. The title of your manuscript should be concise, specific and relevant.

The manuscript has an excellent structure and description. The overall paper is organized and well-written. The literature reviews are insightful and informative.

Unfortunately, figures 2 and 3 had the same explanation – you must verify it.

The conclusion briefly outlines the take-home message and the lessons learned. Conclusions don’t usually have references unless you come up with a ‘punchy’ quote from someone special as a final word. It is recommended not to introduce new bibliographic references in this section because you need more space to discuss them sufficiently.

I congratulate all the authors for their efforts.

Author Response

Thank you for giving us the opportunity to submit a revised draft of our manuscript titled “Identification of a rare Androgen Receptor mutation leading to Complete Androgen Insensitivity Syndrome. The importance of timely diagnosis, optimal therapeutic interventions, and multidisciplinary management.” to Children. We appreciate the time and effort that you and the reviewers have dedicated to providing us with your insightful comments and suggestions, which, hopefully, have significantly improved the manuscript.

We have highlighted the changes within the manuscript using the “Track Changes” function, as you suggested.

Here is detailed answer to the comments.

Yours sincerely,

Dimitris T. Papadimitriou and Maria Papagianni, and on behalf of all our co-authors.

Comments from reviewer 2

  • Comment 1: The title needs to change. The title of your manuscript should be concise, specific and relevant.

Response: Thank you for your comment. We have shortened and modified the title accordingly.

  • Comment 2: Unfortunately, figures 2 and 3 had the same explanation – you must verify it.

Response: Thank you for your comment. We changed the wrong explanation of the figure 3.

  • Comment 3: The conclusion briefly outlines the take-home message and the lessons learned. Conclusions don’t usually have references unless you come up with a ‘punchy’ quote from someone special as a final word. It is recommended not to introduce new bibliographic references in this section because you need more space to discuss them sufficiently.

Response: Thank you for your comment. We moved the discussion regarding the psychological impact of CAIS and the metabolic consequences to the discussion section (lines 358-363), and removed the information about the in-vitro evaluation of the pathogenicity of the AR (lines 370-371).
